# EEG Correlates of Old/New Discrimination Performance Involving Abstract Figures and Non-Words

**DOI:** 10.3390/brainsci11060719

**Published:** 2021-05-28

**Authors:** Monika Toth, Anke Sambeth, Arjan Blokland

**Affiliations:** Department of Neuropsychology and Psychopharmacology, Maastricht University, 6229 ER Maastricht, The Netherlands; anke.sambeth@maastrichtuniversity.nl (A.S.); a.blokland@maastrichtuniversity.nl (A.B.)

**Keywords:** discrimination, recognition memory, EEG old/new effects, abstract figures, non-words

## Abstract

The processing of pre-experimentally unfamiliar stimuli such as abstract figures and non-words is poorly understood. Here, we considered the role of memory strength in the discrimination process of such stimuli using a three-phase old/new recognition memory paradigm. Memory strength was manipulated as a function of the levels of processing (deep vs. shallow) and repetition. Behavioral results were matched to brain responses using EEG. We found that correct identification of the new abstract figures and non-words was superior to old item recognition when they were merely studied without repetition, but not when they were semantically processed or drawn. EEG results indicated that successful new item identification was marked by a combination of the absence of familiarity (N400) and recollection (P600) for the studied figures. For both the abstract figures and the non-words, the parietal P600 was found to differentiate between the old and new items (late old/new effects). The present study extends current knowledge on the processing of pre-experimentally unfamiliar figurative and verbal stimuli by showing that their discrimination depends on experimentally induced memory strength and that the underlying brain processes differ. Nevertheless, the P600, similar to pre-experimentally familiar figures and words, likely reflects improved recognition memory of meaningless pictorial and verbal items.

## 1. Introduction

An efficient memory system requires the ability to detect and incorporate new information and to readily retrieve familiar information [1,2,3,4]. Information-processing theories define recognition as a predicament of discrimination between what is known (familiar) and what is not known (new) [5,6,7,8]. Discrimination performance is typically tested with paradigms that assess recognition memory. In such paradigms, participants are required to recognize previously studied stimuli (e.g., typically well-known and meaningful figures and words) as old correctly and to identify previously not seen items as new [9]. The recognition/discrimination process generates a cue, which induces either a sense of familiarity or unfamiliarity (novelty) [8,10]. The decision process determines how we respond: whether a stimulus is judged as old or new.

While the processing of meaningful and pre-experimentally familiar stimuli is well-researched, less is known about how the brain recognizes and discriminates pre-experimentally unfamiliar and meaningless items. These stimuli can be characterized as previously not seen combinations of simple shapes (abstract figures) and letters (non-words). Knowing how the brain processes and discriminates such stimuli can significantly improve our understanding of visual, lexical, and orthographic memory processing while contributing to establishing relevant cognitive models such as age-related memory impairment. As a result, research in this field could aid patients with orthographic dyslexia, lexical deficiencies, or memory impairment, especially if behavioral and EEG results can be combined with brain–computer interface (BCI) approaches [11,12,13]. Therefore, the current work investigated the behavioral and electrophysiological processing of pre-experimentally unfamiliar figurative and verbal items.

Behavioral data indicate that old/new discrimination at least partly depends on the assessment of the strength of a particular memory. Memory strength is assumed to vary on a continuum ranging from weak to strong, which underlies the subjective perception of stimulus familiarity [8,14]. In other words, the stronger the stimulus memory, the more robust the sense of familiarity appears, and the more likely it is that a stimulus is judged ‘old’ rather than ‘new’.

Previous studies have shown that memory strength, the ease by which a particular memory can be recalled or recognized, can effectively be manipulated as a function of repetition [5] or by using levels of processing (LOP) [15]. Repetition is known to strengthen memory by increasing the subjective sense of familiarity resulting from re-encoding of a particular memory trace [16,17]. The LOP theory predicts that deep (e.g., meaning-extraction, pattern recognition, activation of prior knowledge) and intermediate processing (e.g., phonetics) lead to superior and faster retrieval when compared to shallow processing (e.g., perceptual analyses, rehearsal) [18,19,20,21]. Such findings are supported by brain imaging studies that have found more positive late parietal P600 amplitudes for the deeply- over the shallowly-encoded stimuli [22,23], which corroborates an episodic memory-related function of this evoked response potential (ERP) [24]. For example, Harris and Cutmore [25] used an auditory word recognition task in which half of the trials made use of shallow encoding (i.e., determine if an auditory word comprised long vowels) and the other half used deep encoding (i.e., incorporate an auditory word into a meaningful sentence). According to their findings, not only were the deeply encoded words more accurately recognized than the shallow, but the related P600 amplitudes of the deeply encoded words were also larger compared to the shallow. Moreover, previous research has shown that mnemonics such as verbal (e.g., rhyming), motor (e.g., drawing), or visual (e.g., imagining) techniques strengthen memory storage and improve subsequent retrieval as a result of deeper processing [26,27,28,29].

In the present study, we considered the role of memory strength in the discrimination process of meaningless abstract figures and non-words using a three-phase old/new recognition memory paradigm. We chose to include both figurative and verbal stimuli since these are known to be processed differently [30]. For instance, pictures are generally remembered better than words (i.e., picture superiority effect), since the former in contrast to the latter, are assumed to be encoded with dual (image and verbal) as opposed to singular labels (verbal), respectively [31,32,33]. In the current experiment, we first familiarized the stimuli using mnemonics in order to induce deep processing (deep memorization): we asked the participants to redraw the abstract figures and to come up with existing rhyme words for the non-words (semantic encoding). In the second phase, participants were asked to merely study the stimuli (shallow memorization). Here, the previously deeply encoded items were shown again in combination with some new items. Finally, an old/new recognition test was applied in which stimuli from the first and second phases were intermixed with new ones. We assessed both recognition accuracy and speed.

According to our knowledge, this is the first study examining the impact of memory strength on the discrimination performance of pre-experimentally unfamiliar items using such a design. It is known that identifying something as new compared to it being old usually takes longer than old item recognition [1,2,34]. Therefore, we assumed that the recognition of the drawn/semantically processed and studied items would be faster compared to the correct identification of the new items. Moreover, we anticipated that the drawn/semantically processed stimuli would be recognized more accurately and faster than the studied ones.

To match the possible effects of new and old item recognition with brain processes, we measured electrophysiological responses during retrieval. Four ERPs of interest were the N200, P300, N400, and P600. Recent research has shown how these different ERPs can be helpful in the context of memory research and BCIs. For example, the P300 has been used to discriminate familiar from unfamiliar faces [35]. Based on the N400, it was possible to infer semantic relatedness of verbal stimuli [36,37], and with the P600, it was possible to differentiate between target and lure audiovisual number stimuli [37].

EEG data on the processing of abstract images are scarce and not well understood. For instance, identification of unusual stimuli in oddball tasks typically elicits larger N200 and P300 amplitudes [38]. Interestingly, O’Hare and Goodwin [39] found that abstract artwork images elicited larger P300 amplitudes than natural ones. At the same time, they failed to detect differences for the N200. Beisteiner and Huter [40] failed to find differences in the P300 amplitude during the recognition of old and new abstract geometric images. Later old/new correlates typically elicit the so-called early (N400) and late (P600) old/new effects, respectively [24]. The fronto-central N400 is postulated to represent the effective use of familiarity to discriminate between old and new items [24,41,42]. As such, early old/new effects are seen as more negative amplitudes for new stimuli that were correctly endorsed as ‘new’ compared to previously seen old stimuli that were recognized as ‘old’ [24,41,42]. Applying this to the current paradigm, more negative amplitudes of the correctly detected new compared to old images would reflect optimal discrimination of these stimuli. The parietal late old/new effect (P600) typically entails larger amplitudes of the pre-exposed compared to the not pre-exposed items [8,24,41]. At the same time, this reflects improved memory performance [30,43]. Therefore, we anticipated a more positive late old/new effect for the drawn and repeated figures that relied on strong memories compared to the new ones. Likewise, we expected to find late old/new effects for the studied figures that relied on weak memories.

Electrophysiological processing of non-words is somewhat better understood. For instance, whereas words affect the N400, non-words influence the P300 [44,45]. The amplitude of the P300 is thought to index context updating as a result of initiated input comparisons to existing representations [38,45,46]. Its amplitude enhances whether a stimulus does not fit or weakly matches any existing representation [46]. Additionally, according to the context-updating hypothesis, increased N200 and decreased P300 amplitudes are indicative of a match to an existing memory template, and the opposite is true if the match is vague [38]. Therefore, semantically processed stimuli should elicit the former pattern resulting from deeper LOP and repetition. In contrast, the studied stimuli should show the latter pattern as a result of shallow LOP without repetition. Considering that the P600 has been linked to successful old/new item discrimination and has been found with words [47,48] and non-words [49], we assumed we would detect late parietal old/new effects. This was expected to show in a larger peak amplitude for the correctly recognized old than new non-words.

Finally, in terms of the LOP manipulation, we expected to detect more positive amplitudes for the deeply- over the shallowly-memorized abstract figures and non-words [22,23].

## 2. Materials and Methods

The Ethical Committee of the Faculty of Psychology and Neuroscience of Maastricht University granted ethical approval for this experiment. Each participant received monetary compensation or research participation credit points. On average, the experiment took 1.5 hour/test-session (Ethical Approval Code: ECP13_02_2012).

### 2.1. Participants

A total of 22 young, healthy participants (eight males) with a mean age of 24 years were recruited by means of advertising. The main inclusion criteria were age (18–30 years) and being fluent in the English or Dutch languages. One participant was excluded from the behavioral analyses because of a technical failure.

### 2.2. Procedure

After signing an informed consent, participants were admitted to the study. Prior to starting the experiment, each participant filled in a demographic questionnaire including information about sex, age, and handedness. At this point, the EEG caps were installed. During the test, stimuli were presented via a computer screen, and participants had to respond on two keys of a response pad. Recognition accuracies and reaction times were recorded.

A memory paradigm with abstract figures and non-words was applied in separate tests (see Figure 1 for an example of the stimuli used). Every participant performed each test phase first with the abstract figures and then with the non-words in order to minimize verbalization of the figurative stimuli. The experiment consisted of three phases (see Figure 2). In phase 1 (deep memorization leading to ‘strong’ memory), participants were familiarized with a series of 30 monosyllabic abstract figures or non-words in separate tests (list 1: L1). Participants were asked to manually redraw the abstract figures on an answer sheet in order to induce deep LOP. They had to mention existing English or Dutch rhyming words for each non-word to induce intermediate LOP. Stimuli were presented for 1 s, and the participants were given 14 s to execute the mnemonic encoding task. If they were ready earlier, they could press a button, and 2 s later, the next stimulus appeared. Stimuli were extracted from previous studies [50,51,52].

During phase 2 (shallow memorization leading to ‘weak’ memory), participants were instructed to remember as many stimuli as possible. In this phase, 60 stimuli (abstract figures or non-words) were used: 30 stimuli from L1 were randomly mixed with 30 new ones (L2). All stimuli were shown for 1 s with an inter-stimulus interval (ISI) of 2 s.

During phase 3, participants were asked to decide if they had seen the presented stimulus in the previous series (L1 and L2) or whether the stimulus was new to them (L3: new, *n* = 30). The 90 non-words or abstract figures were presented for a duration of 1 s, or less in the case of faster button press; the ISI was 2.5 s. Participants had to press the corresponding buttons (‘old’ for L1 and L2, or ‘new’ for L3 stimuli) on a response box as quickly and accurately as possible.

A filler paper-and-pencil task of 10 min and another non-verbal task were given between phase 2 and 3. The filler task consisted of the localization of number sequences, vertically or horizontally placed within a field of numbers (10 min). The other task consisted of watching a silent cartoon while auditory stimuli were presented (10 min).

EEG was recorded simultaneously with the behavioral testing. Recordings were carried out with a standard EEG apparatus using 32 electrodes, placed according to the 10/20 standard international placement of electrodes. Eye movements were monitored through electrooculograms, with electrodes placed above and below the left eye as well as at the corner of the left and right eyes. The ground electrode was placed at FPz. Reference electrodes were applied at the left and right mastoid bones. Data were sampled at 512 Hz and filtered between 0.05 and 100 Hz during acquisition. After the recording, data were pre-processed using Brain Vision Analyzer 2 (Brain Products, Munich) including filtering (1–30 Hz), ocular correction, segmentation, baseline correction (−100 to 0 ms), artifact rejection (exclusion of trials if signal exceeded +100 or −100 μV), and calculation of averages and grand averages by averaging the responses between 100 ms before and 1000 ms after stimulus onset for the correct responses of each stimulus type separately. Sequentially, the data were clustered into frontal (F3, F4, and Fz), central (C3, C4, and Cz), and posterior (P3, P4, and Pz) sections [24]. Based on visual inspection of the grand averages, the N200, P300, N400, and P600 peak amplitudes and latencies were determined using the time windows presented in Table 1. Next, as suggested by Luck [53], we applied the local peak amplitude detection method available in Brain Vision Analyzer 2.

### 2.3. Data Analysis

Before analysis, all data were evaluated for having normal distribution and homogeneity of variance. Additionally, raw data were checked for outliers. Outlier values were replaced with their regression estimates produced by the missing value analyses (IBM SPSS Statistics for Macintosh, Version 27.0. Armonk, NY, USA: IBM Corporation). Additionally, due to technical issues, 1–2 responses per participant were missing (e.g., the button press was not recorded). In these cases, values were replaced with their regression estimates. Effect sizes are reported based on partial eta-squared (ηp^2^) data. Furthermore, Mauchly’s test of sphericity was applied. In case the assumption of sphericity was violated, a Greenhouse–Geisser correction was used. In all cases, degrees of freedom of assumed sphericity were reported. Post-hoc comparisons and simple effects were investigated using paired-samples t-tests, applying adjustments for multiple comparisons; the observed p-values were multiplied by the number of comparisons, which was tested against the set significance level of 0.05.

For the behavioral data, signal detection theory (SDT) was applied in order to investigate the discrimination performance [5,6,7,54]. Discrimination accuracy was defined as the ability to distinguish the different types of stimuli (drawn/semantically processed, studied, and new). Correct responses included an ‘old’ response to the drawn/semantically processed items, and the studied stimuli, and a ‘new’ response to the new items. Incorrect responses involved a ‘new’ response to the drawn/semantically processed items and the studied stimuli and an ‘old’ response to the new stimuli. See Table 2 for an overview.

Given the memory strength manipulation in the current design (deep memorization, shallow memorization, and recognition), the correct response rates, being hit rates (HR) for the drawn/semantically processed, and the studied items and correct rejection rates (CRR) for the new, were used to evaluate the discrimination accuracy. Furthermore, in order to investigate discriminability, non-parametric A’ statistics were computed for the drawn/semantically processed and the studied stimuli using Equation (1) or Equation (2) (see below) [6,55]. A′ varies from 0 to 1, with 0.5 indicating chance performance. Higher values are indicative of improved performance [6,55].
(1)$$A\prime =0.5+\frac{\left(HR-FAR\right)\left(1+HR-FAR\right)}{4HR\left(1-FAR\right)},\;\mathrm{if}\;HR\geq FAR$$
(2)$$A\prime =0.5-\frac{\left(HR-FAR\right)\left(1+HR-FAR\right)}{4HR\left(1-FAR\right)},\;\mathrm{if}\;HR<FAR$$

A’: discriminability index, HR: hit rate, FAR: false alarm rate.

During recognition, the a priori probabilities of old and new items and the quality of the match between a test item and the memory for studied items can influence the bias parameter [6,45]. Such a model does not fit the current paradigm due to the memory strength manipulation used and the equivalent proportion and intended comparison of the strong (*n* = 30), weak (*n* = 30), and new items (*n* = 30) [44]. After all, the final proportion of ‘old’ and ‘new’ responses was 2:1. Therefore, we calculated the total amount of ‘old’ (H + FA) and ‘new’ (M + CR) responses given by the participants. This was done to examine whether there was a preference for either the ‘old’ or ‘new’ responses. Results were compared using paired samples t-tests with Bonferroni corrections.

RT data of the hits were also evaluated. To be able to use parametric tests, RT-s were transformed into |log(1/RT)| to obtain a normal distribution of the data [46]. Moreover, the median RT data are reported as central tendency parameters, together with the corresponding first and third interquartile ranges [47].

Statistical analysis was conducted using SPSS 27.0. A repeated-measures analysis of variance (ANOVA) was used to investigate the discrimination accuracy scores and RT-s for the different stimuli in the different categories as assessed in phase 3. The within-subject variables were stimulus category (abstract figures and non-words) and stimulus type (drawn/semantically processed, studied, and new items).

For the EEG analysis, a within-subjects repeated-measures analysis of variance (ANOVA) was conducted to investigate the amplitude and latency data during recognition (phase 3). The model included the following factors within each stimulus category: stimulus type with three levels (drawn/semantically processed, studied, and new items) and location with three levels (frontal, central, and parietal). Since the aim of the study was to investigate discrimination performance, the main effects and interactions involving stimulus type (amplitudes and latency data) are described in detail.

## 3. Results

### 3.1. Behavioral

Although there was an unequal number of old-responses over new-responses (2:1), we found that there was no response bias (see Table 3). The mean signal-detection parameter estimates are displayed in Table 4.

#### 3.1.1. Abstract Figures

With respect to the accuracy scores (HR and CRR) of the abstract figures, the ANOVA yielded a significant main effect of stimulus type [*F*(2.40) = 70.24, ηp^2^ = 0.78, *p* < 0.001; see Figure 3A and Table 4]. Post-hoc tests revealed that the drawn stimuli were recognized more accurately than the studied (*p* < 0.001) and the new (*p* < 0.027). Additionally, more new stimuli were correctly identified compared to the studied items (*p* < 0.001).

The analyses performed on the A’ scores revealed that the drawn abstract figures resulted in improved discriminability compared to the studied [*F*(1.20) = 278.10, ηp^2^ = 0.93, *p* < 0.001; see Table 4].

When analyzing the reaction time performance in the session with the abstract figures, a main effect of stimulus type was found [*F*(2.40) = 36.68, ηp^2^ = 0.65, *p* < 0.001; see Table 5]. Post-hoc tests showed that response times were significantly faster to the drawn stimuli compared to the studied (*p* < 0.001) and new items (*p* < 0.001). No such difference was found between the studied and new abstract figures (*p* > 0.999).

#### 3.1.2. Non-Words

When investigating the accuracy performance (HR and CRR) of the non-words, the analyses revealed a significant main effect of stimulus type [*F*(2.19) = 31.07, ηp^2^ = 0.77, *p* < 0.001; see Figure 3B and Table 4]. Post-hoc tests revealed that the semantically processed stimuli were recognized more accurately than the studied (*p* < 0.001). Additionally, more new stimuli were correctly identified compared to the studied (*p* < 0.001). No such difference was found between the semantically processed and new non-words (*p* > 0.849).

The analyses of the A’ scores revealed that it was easier to discriminate the semantically processed non-words than the studied [*F*(1.20) = 40.05, ηp^2^ = 0.67, *p* < 0.001; see Table 4].

As for the reaction times, stimulus type yielded a significant main effect [*F*(2.19) = 6.45, ηp^2^ = 0.40, *p* < 0.007; see Table 5]. Post-hoc tests showed that response times were significantly faster to the studied than the new non-words (*p* < 0.015). No such difference was found between the semantically processed and new items (*p* > 0.069), and the semantically processed and studied non-words (*p* > 0.999).

### 3.2. EEG

#### 3.2.1. Abstract Figures

N200

The analyses revealed a significant stimulus type × location interaction [*F*(4.84) = 3.98, ηp^2^ = 0.16, *p* < 0.014; see Figure 4A–C]. Simple effects analyses showed that the amplitude related to the studied abstract figures was more negative than that related to the drawn figures at the posterior location (*p* < 0.039). The rest of the combinations were found to be insignificant (all *p*-values > 0.105). Stimulus type did not reveal a main effect [*F*(2.20) = 1.31, ηp^2^ = 0.12, *p* > 0.292]. The latency of the N200 was not affected by stimulus type or location [all associated *F*-values < 1.85, ηp^2^ < 0.17, *p*-values > 0.183].

P300

When analyzing the amplitude data of the P300, a significant stimulus type x location interaction was detected [*F*(4.84) = 5.49, ηp^2^ = 0.21, *p* < 0.003; see Figure 4A–C]. However, simple effects analyses did not reveal any significant differences (all *p*-values > 0.168). Stimulus type did not reveal a main effect [*F*(2.20) = 0.74, ηp^2^ = 0.10, *p* > 0.492].

As for the latency data, neither stimulus type nor the interaction between stimulus type and location was statistically meaningful [all associated *F*-values < 1.42, ηp^2^ < 0.24, *p*-values > 0.267].

N400

According to the analyses, the stimulus type x location interaction was significant [*F*(4.84) = 6.93, ηp^2^ = 0.25, *p* < 0.001; see Figure 4A–C and Figure 5A]. Simple effects analyses revealed that the amplitude related to the new abstract figures was more negative than that representing the drawn items at the frontal (*p* < 0.003), central (*p* < 0.003), and posterior locations (*p* < 0.003). Additionally, the N400 amplitude related to the studied abstract figures was more negative compared to the drawn at the central ( *p* < 0.001) and posterior (*p* < 0.001) but not the frontal location (*p* > 0.999). The rest of the combinations were not significant (all *p*-values > 0.999).

As for the latency, neither the interaction stimulus type x location nor stimulus type was significant [all associated *F*-values < 2.70, ηp^2^ < 0.38, *p*-values > 0.064].

P600

The analyses showed a significant stimulus type × location interaction [*F*(4.84) = 13.68, ηp^2^ = 0.40, *p* < 0.001; see Figure 4A–C and Figure 5C]. Simple effects analyses revealed that the amplitude related to the drawn abstract figures was larger than that of the studied at the central (*p* < 0.003) and posterior locations (*p* < 0.003). Additionally, the P600 in response to the drawn items was larger compared to the new at the central (*p* < 0.033) and posterior (*p* < 0.006) locations. No other differences were detected (all *p*-values < 0.968).

As for the latency, there was a significant main effect of stimulus type [*F*(2,20) = 7.47, ηp^2^ = 0.43, *p* < 0.004; see Figure 4A–C and Figure 5D]. Post-hoc tests showed that the P600 latency related to the new items was shorter compared to the studied (*p* < 0.018). No further differences were found (all associated *p*-values > 0.100). The interaction between stimulus type and location was not significant [*F*(4.84) = 0.67, ηp^2^ = 0.03, *p* > 0.546].

#### 3.2.2. Non-Words

N200

The analyses did not reveal any significant effects for the amplitudes [all associated *F*-values < 2.33, ηp^2^ < 0.19, *p* > 0.123] or the latencies [all associated *F*-values < 2.42, ηp^2^ < 0.20, *p* > 0.114].

P300

The amplitudes were dissimilar for the different stimulus types [*F*(2.42) = 4.44, ηp^2^ = 0.17, *p* < 0.029]. Post-hoc tests revealed that the amplitude related to the studied non-words were marginally larger than those related to the new (*p* = 0.053). No further statistically meaningful differences were found (all *p*-values > 0.165). The stimulus type x location interaction was not significant [*F*(4.84) = 1.34, ηp^2^ = 0.06, *p* > 0.268].

The latency analyses revealed a significant effect of stimulus type [*F*(2.42) = 9.58, ηp^2^ = 0.31, *p* < 0.001; see Figure 6A–C]. Post-hoc tests revealed that the P300 in response to the semantically processed stimuli peaked later compared to the studied (*p* < 0.017) and the new ones (*p* < 0.006). No such difference was found between the studied and semantically processed non-words (*p* > 0.282). Finally, neither stimulus type nor the interaction term stimulus type x location was significant [all associated *F*-values < 1.22, ηp^2^ < 0.11, *p* > 0.317].

N400

The analyses did not reveal any significant effects for the amplitudes [all associated *F*-values < 2.65, ηp^2^ < 0.12, *p* > 0.099] or the latencies [all associated *F*-values < 1.41, ηp^2^ < 0.06, *p* > 0.250].

P600

The analyses showed a significant stimulus type x location interaction [*F*(4.84) = 8.10, ηp^2^ = 0.28, *p* < 0.001; see Figure 6A–C and Figure 7A]. Simple effects analyses revealed that the P600 amplitude related to the new non-words was smaller than that related to the semantically processed (*p* < 0.009) and the studied non-words (*p* < 0.042) at the posterior location. No other differences were detected (all *p*-values < 0.258). Stimulus type did not affect the amplitudes [*F*(2.42) = 2.05, ηp^2^ = 0.09, *p* > 0.148].

As for the latency, there was a significant main effect of stimulus type [*F*(2.20) = 3.97, ηp^2^ = 0.28, *p* < 0.035; see Figure 6A–C and Figure 7B]. Post-hoc tests showed that the P600 latency related to the strong items was shorter compared to the new (*p* < 0.027). No other differences were found (all associated *p*-values > 0.482). Finally, the interaction between stimulus type and location was not significant [*F*(4.84) = 0.67, ηp^2^ = 0.03, *p* > 0.546].

## 4. Discussion

In the current study, we examined the role of memory strength in the old/new discrimination process involving abstract figures and non-words. To account for the effect of memory strength, we manipulated LOP and used repetition. The present findings, using pre-experimentally unfamiliar and meaningless abstract figures and non-words, are in line with the notion that discrimination performance depends on how strong a particular newly formed memory is. Namely, correct identification of the new abstract figures and non-words was superior to the recognition of the old items when they were merely studied, and thus, relied on weaker memories (shallowly memorized and not repeated). This was not the case when the stimuli were drawn or semantically processed, and thus, relied on stronger memories (deeply memorized and repeated). Additionally, despite the unequal proportion of old and new items (2:1) during the recognition phase, we did not detect an ‘old’ response bias for either the abstract figures or the non-words.

The EEG analysis indicated that the amplitude of the N200 was more negative during the processing of the studied than the drawn figures at the posterior location. The N400 was more negative for the new than the strong and the studied figures at the fronto-central FN400 (early old/new effect) and at the posterior location. In addition, the strong figures elicited a larger P600 amplitude in comparison to the studied and new stimuli at the centro-parietal locations (late old/new effect). Furthermore, the latency related to the processing of the new figures was shorter than that related to the studied. The ERPs of the non-words were not as clear and straightforward as seen with the abstract figures. The semantically processed non-words elicited larger P600 amplitudes than the studied or the new ones, representing a late old/new effect. There was also a large variety in the latencies. The only significant latency effect was detected for the P600. Namely, the latency associated with the processing of the semantically processed non-words was longer than that of the new. Below, we discuss how these ERP findings can explain discrimination performance.

### 4.1. Abstract Figures

As expected, we found that stronger memories achieved by deeper mnemonic LOP and repetition improved recognition accuracy and decreased the RT-s of the drawn (deeply memorized and repeated) as opposed to the studied (shallowly memorized) figures. These findings align with the notion that deeper LOP improves recognition and discrimination performance [26,27,28]. The current data also show that it was likely to be more difficult to correctly discriminate the studied abstract figures as being old or new since their discriminability indexes (A’) were lower than those of the drawn. This result is consistent with previous research [56,57].

Furthermore, correct new item identification was superior when abstract image memory was weak (studied items memorized shallowly) but not when the memory was strong (drawn abstract figures that were deeply memorized and repeated). This differential effect of deep vs. shallow LOP and repetition vs. no repetition on the recognition performance indicates a mediating effect of memory strength. Prior research on discrimination performance involving well-known stimuli has typically shown that new items are processed more accurately than old ones [1,2,3,4]. However, when new abstract memories are formed, it seems that familiar (old) abstract figurative items that rely on strong are processed more accurately than the unfamiliar (new) ones. Moreover, in line with previous findings, the drawn figures having the strongest memory were judged ‘old’ correctly the fastest [1,2,34].

When looking at the brain activity data, we found that the studied figures’ processing elicited a more negative N200 amplitude than the drawn ones at the posterior location. In oddball tasks, the N200 amplitude increases in response to an unpredictable stimulus in a pool of frequently repeated items [38]. The N200 has also been linked to a mismatch between an existing memory template and the test stimulus, thus perceiving something as new [58]. Moreover, this ERP is assumed to be sensitive to perceptual novelty in that a single repetition attenuates its amplitude [59]. Thus, on the one hand, it could be that the weakly embedded abstract figures were regarded as relatively new, which explains why their N200 amplitudes increased. On the other hand, the drawn figures were repeated, which could account for their attenuated amplitudes.

Furthermore, in agreement with Beisteiner and Huter [40], and in contrast to O’Hare and Goodwin [39], we failed to find differences for the P300. This can be explained by the nature of the images and experimental designs used in these experiments. Namely, we used simple black and white abstract line drawings in an old/new recognition memory paradigm. Likewise, Beisteiner and Huter [40] used black and white abstract geometric images and tested old/new recognition. O’Hare and Goodwin [39] used rather complex and colorful abstract artwork and a viewing task accompanied with discomfort judgements. Thus, while brain responses might differ depending on the complexity of the abstract imagery, it seems that old/new discrimination of simple abstract figures neither affects the P300 amplitude nor the latency.

However, the current data showed a greater negative deflection of the N400 for the new compared to the drawn, and the studied compared to the drawn abstract figures over the fronto-central and posterior regions. This finding supports the notion that the N400 reflects familiarity-based discrimination and correct identification of new or fragilely memorized items [24,42]. The more posterior N400 is typically related to semantic processing [44,60]. Thus, the finding that the drawn abstract figures evoked the largest amplitudes at the more posterior locations could indicate that the participants tried to use verbalization during the processing of these stimuli. However, we did not ask which memory strategies the participants used.

The P600 amplitudes of the drawn figures were larger compared to the new and studied ones. Since this ERP component is known to reflect recollection memory and successful old/new discrimination [24,41,42,44,60], this could indicate that the drawn figures and their encoding context were recollected better, probably as a function of the memory strength manipulation. Our behavioral results also supported such a notion.

To summarize, the superior discrimination performance for the new figures can be explained by a more negative N400 amplitude. Larger P600 peaks can explain the improved recognition of the deeply memorized and repeated (drawn) figures. The N400 amplitude related to the processing of the studied figures was comparable to the new ones and showed a P600 wave similar to them, which elucidates their poorer discrimination performance. Thus, accurate discrimination performance of the abstract figures can be explained by a combination of effective use of familiarity (N400) and recollection (P600) type memory for the deeply encoded and repeated items.

### 4.2. Non-Words

The behavioral findings regarding the advantages of deeper mnemonic encoding and repetition were very similar for the non-words and the abstract figures. Thus, semantically processed non-words were endorsed correctly as ‘old’ just as well as the new ones were identified as ‘new’. Additionally, deeper LOP and repetition resulted in improved accuracy compared to shallow memorization without repetition. Our findings on the LOP effect with the non-words were comparable to those detected with actual words by Gardiner [15]. Half of the participants encoded well-known words in his experiment by writing down a rhyming word (intermediate encoding); the other half had to come up with a semantic associate for each word (deep encoding). After a delay, an old/new recognition test was applied. Accuracy scores in the deep condition were significantly higher than in the intermediate condition. Thus, it can be said that deeper compared to more shallow LOP does not only improve the discrimination of words, but also of non-words.

The reaction times in response to the new non-words were slower than to the studied. Similar findings have previously been reported with non-words [49], whereas the opposite was found with words [47]. It could be that the nonsense letter combinations of the new non-words in the current experiment were simply more challenging to process due to some sort of confusion or conflict with existing verbal knowledge, which led to more extended processing. In other words, it might be difficult for the brain to make sense of random letter strings upon a first encounter.

Although the behavioral results were similar to the abstract figures, EEG processing appeared to be different due to the diverse nature of the stimuli. Notably, the ERP components visible for the non-words appeared to be less clear than those of the abstract figures or components typically observed with words [47,48]. Similar visual data can be found in the experiment by Otten and Sveen [44], who investigated the electrophysiological processing of words and non-words during encoding and recognition. The most feasible explanation for the qualitatively worse representation of the word vs. non-word ERPs could be that only the former has meaning. This enables us to obtain clearer signals during word over non-word processing since actual words make sense.

In contrast to our expectations, we failed to detect significant N200 and P300 effects or find support for the context updating hypothesis with the non-words [38]. We also could not detect differences involving the N400 amplitude or latency. This can be explained by the non-words being semantically incorrect [60]. Indeed, other studies using meaningless stimuli also did not find effects on the N400 [61,62]. However, this contrasts with our own findings regarding the abstract figures, which were also meaningless.

As seen with the abstract figures, the P600 amplitude of the non-words was also different for the semantically processed, studied, and new stimuli. Namely, while the semantically processed and the studied non-words elicited similar ERPs, they were both larger than those of the new. Such late old/new effects align with previous research showing that the later posterior P600 reflects recognition memory and effective old/new discrimination [24,41,42,44,60]. As such, our results agreed with those by Fjell and Walhovd [49], who used both word and non-word stimuli in a two-phase EEG memory experiment. As in the current paradigm, their first phase was meant to induce deeper LOP utilizing a lexical discrimination task (word or non-word), followed by an old/new recognition test. Their finding revealed a clear parietal P600 peak for both the old words and the non-words, which the authors attributed to the quality and quantity of the recollected episodic memory. Thus, it seems that the deep memorization task applied with the semantically processed non-words, similar to the drawn abstract figures, contributed to memory recollection. This notion aligns with our finding, according to which the P600 latency associated with the new non-words was shorter than that related to the semantically processed ones. Accordingly, processing of the semantically processed items might have been more demanding due to recollection of contextual details (pairs with the rhyme words during deep memorization). Our findings also indicate that memory formation and retrieval of the semantically processed and the studied items were likely successful in the brain. Nonetheless, the recognition accuracy of the studied stimuli was worse than that of the semantically processed and new non-words.

## 5. Conclusions

The present study shows that the discrimination performance of pre-experimentally unfamiliar abstract figures and non-words depends on memory strength induced by deeper LOP and repetition. Specifically, new abstract figures and non-words were more accurately identified than the old ones that relied on weak memories. In contrast to the stimuli relying on weak memories, discrimination of the drawn/semantically processes and new stimuli was improved when their memory was strong or when the memory was absent, respectively. By using pre-experimentally unfamiliar abstract figures and non-word stimuli, the current experiment largely controlled for retroactive interference.

The EEG results of the abstract figures indicated that the N400 effect discriminates between the new and drawn items relying on strong memories, whereas the P600 effect discriminates between the recollected drawn and non-recollected items (studied items with weak memory and new stimuli). The late parietal P600 likely differentiates between old and new non-words.

In closing, the presented findings fill a gap in the research by providing new insight into how the brain processes and discriminates abstract figures and non-words and what role memory strength has in these processes. Essentially, our findings improve the understanding of such visual and lexical memory processing that does not rely on semantic knowledge. Moreover, we could identify a pattern of brain activity that defines accurate discrimination performance of such items, which could be useful in developing appropriate BCI technologies.

To further investigate the claims presented in this work, future research could implement spectral power analyses. Moreover, the present paradigm could be extended with the application of confidence judgments and questionnaires that could reveal which specific strategies the participants used during memorization. An intriguing follow-up study could examine the relation between recognition and recall using an extended version of the current paradigm. Finally, it would be interesting to define the electrophysiological patterns of pre-experimentally unfamiliar old/new item recognition and combine these results with BCI technology.

## Figures and Tables

**Figure 1 brainsci-11-00719-f001:**
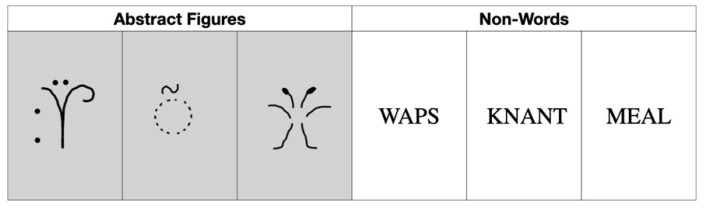
Examples of the stimuli used.

**Figure 2 brainsci-11-00719-f002:**
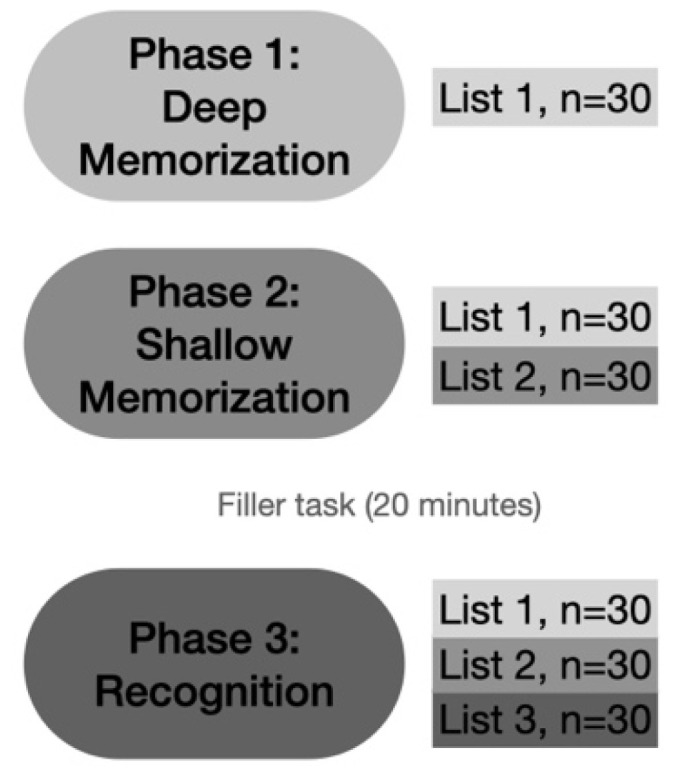
Schematic overview of the experimental design. Phase 1: deep memorization with the pre-experimentally unfamiliar abstract figures and non-words in separate tests using a mnemonic encoding task (redrawing the abstract figures and mentioning rhyming words for the non-words). The 30 stimuli used here form List 1 (drawn/semantically processed stimuli). Phase 2: shallow memorization with the instruction to remember as many stimuli as possible. This phase contained items from List 1 and 30 new ones (List 2, studied stimuli). Phase 3: recognition of the stimuli including List 1, List 2, and 30 new (List 3). *n*: number of stimuli presented.

**Figure 3 brainsci-11-00719-f003:**
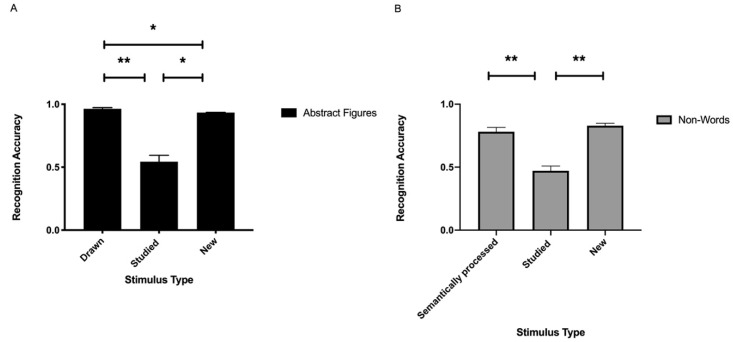
Recognition accuracy performance of the abstract figures (**A**) and the non-words (**B**) according to stimulus type. The bars represent the means of the hit rates of the drawn/semantically processed and the studied items, and the means of the correct rejection rates for the new items. Stimulus type effects: **: *p* < 0.001, *: *p* < 0.05.

**Figure 4 brainsci-11-00719-f004:**
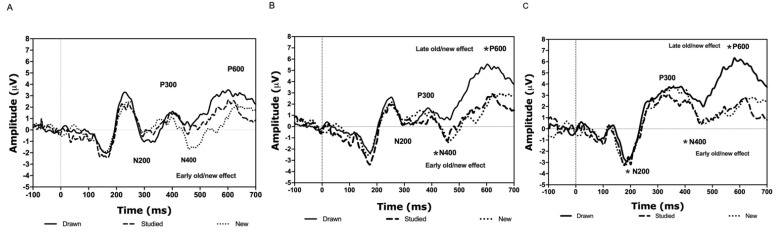
Grand averages across participants for the N200, P300, N400 and P600 during the recognition of the abstract figures according to stimulus type at the frontal (**A**), central (**B**), and posterior (**C**) electrode location clusters. Stimulus type effects: *: *p* < 0.05.

**Figure 5 brainsci-11-00719-f005:**
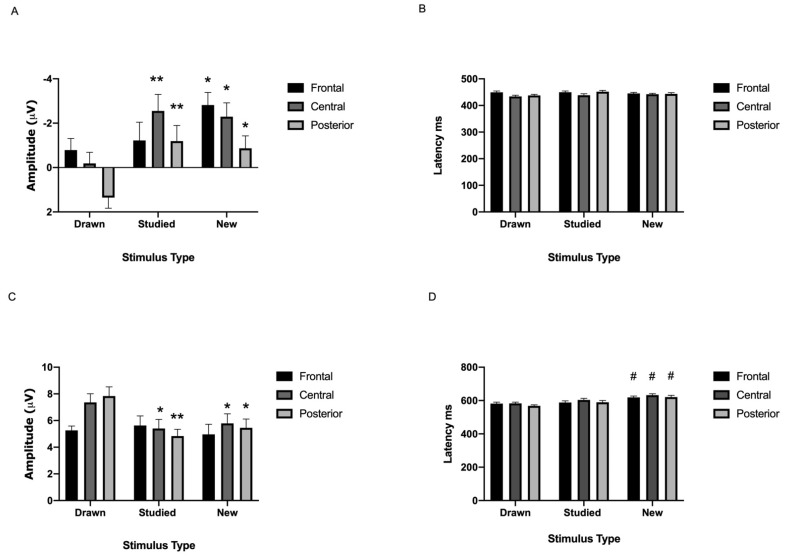
Means of the N400 amplitude (**A**) and latency (**B**), and the P600 amplitude (**C**) and latency (**D**) during the recognition of the abstract figures. Stimulus type effects: different from the drawn items: **: *p* < 0.001, *: *p* < 0.05, different from the studied items: ^#^: *p* < 0.05.

**Figure 6 brainsci-11-00719-f006:**
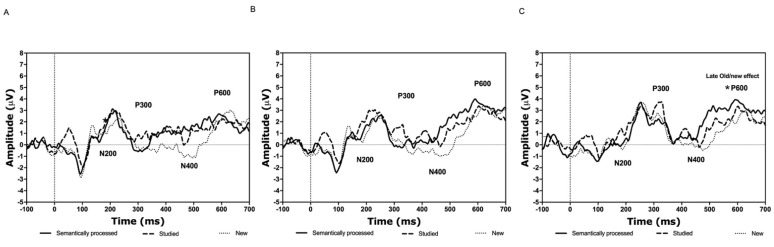
Grand averages across participants for the N200, P300, N400 and P600 during the recognition of the non-words according to stimulus type at the frontal (**A**), central (**B**), and posterior (**C**) electrode location clusters. Stimulus type effects: *: *p* < 0.05.

**Figure 7 brainsci-11-00719-f007:**
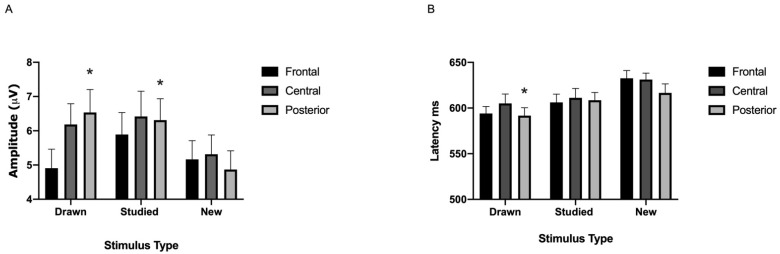
Means of the P600 amplitude (**A**) and latency (**B**) during the recognition of the non-words. Stimulus type effects: different from the drawn items: *: *p* < 0.05.

**Table 1 brainsci-11-00719-t001:** Overview of the time windows of the analyzed ERP components during recognition (in milliseconds).

Stimulus Category/ERP	N200	P300	N400	P600
Abstract Figures	200–320	300–420	400–500	500–700
Non-words	150–250	250–400	400–500	500–700

**Table 2 brainsci-11-00719-t002:** Overview of the different types of responses as a function of stimulus type.

	Stimulus Type	Response
Hit (H)	Drawn or semantically processed/Studied	‘Old’
Miss (M)	Drawn or semantically processed/Studied	‘New’
Correct Rejection (CR)	New	‘New’
False Alarm (FA)	New	‘Old’
Hit Rate (HR)	Drawn or semantically processed/Studied	H/(H + M)
Correct Rejection Rate (CRR)	New	CR/(CR + FA)

**Table 3 brainsci-11-00719-t003:** The total number of old and new responses during the recognition test. Data represent the means (SEM) of the total ‘old’ and ‘new’ responses and the corresponding % compared to the 90 items/stimulus category (abstract figures and non-words), and the t-statistics.

	Abstract Figures		Non-Words	
Old responses	47.24 (2.04)	52%	43.10 (1.95)	48%
New responses	42.10 (2.10)	47%	46.52 (1.97)	52%
Paired samples *t*-test	t(20) = 1.25, *p* > 0.450		t(20) = 0.88, *p* > 0.784	

**Table 4 brainsci-11-00719-t004:** Means (SEMs) of the signal-detection measures of the recognition performance of the abstract figures and non-words according to stimulus type (drawn/semantically processed, studied, and new).

Stimulus Type	Parameters	Abstract Figures	Non-Words
Drawn/Semantically processed	HR	0.96 (0.01) **^,#^	0.78 (0.04) **
	A’	0.88 (0.01) **	0.67 (0.02) **
Studied	HR	0.54 (0.05)	0.47 (0.04)
	A’	0.62 (0.02)	0.55 (0.01)
New	CRR	0.93 (0.01) **	0.83 (0.02) **

HR: hit rate, CRR: correct rejection rate, A’: discriminability index. Different from studied: **: *p* < 0.001, difference between the drawn and new items: ^#^: *p* < 0.05.

**Table 5 brainsci-11-00719-t005:** Median reaction times (middle 50% range; in milliseconds in response to the abstract figures and non-words (the drawn/semantically processed, studied, and new) and their corresponding first and third interquartile ranges.

Stimulus Type	Abstract Figures	Non-Words
Drawn/Semantically processed	649 (612–739)	643 (619–667)
Studied	794 ** (714–957)	642 ^#^ (575–695)
New	810 ** (744–918)	699 (644–739)

Different from drawn/semantically processed (stimulus type effects): different from drawn: **: *p* < 0.001, different from studied: ^#^: *p* < 0.05.

## Data Availability

The data and materials for the experiment are available from the corresponding author on request.

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
