# Peer review of "EEG Correlates of Old/New Discrimination Performance Involving Abstract Figures and Non-Words"

_brainsci, 2021, doi:10.3390/brainsci11060719_

Round 1

Reviewer 1 Report

Undoubtedly, the processing of pre-experimentally unfamiliar stimuli, such as abstract figures and non-words, is poorly understood. Authors considered the role of memory strength in the discrimination process of such stimuli using a three-phase old/new recognition memory paradigm.

My comments to the article are as follows:

- I propose to extend the Introduction by referring to implementation examples of EEG signal use, for example in BCI technology: Using the Raspberry PI2 Module and the Brain-Computer Technology for Controlling a Mobile Vehicle, Automation 2019: Progress In Automation, Robotics And Measurement Techniques, Advances in Intelligent Systems and Computing from Springer, 2020. In addition, I propose to refer to the topic of EEG signal source location, including, for example, the publication: Characteristics of Question of Blind Source Separation Using Moore-Penrose Pseudoinversion for Reconstruction of EEG Signal, Automation 2017: Innovations In Automation, Robotics And Measurement Techniques, Book Series: Advances in Intelligent Systems and Computing from Springer, 2017. The extension of information in the field of methods of acquisition and archiving of signals related to the work of the human brain is also under consideration. For example, you can refer to: Methods of Acquisition, Archiving and Biomedical Data Analysis of Brain Functioning, Biomedical Engineering And Neuroscience, Book Series: Advances in Intelligent Systems and Computing from Springer, 2018. Your bibliography will therefore be updated.

- Please provide arguments on what basis did you identify the group of people for the study?

- Please provide arguments on what basis did you choose the statistical method ANOVA?

- I propose to move Figures, tables and schemes from point 3.3 to the appropriate points in the content of the article.

- Moreover, Figures 6a, 6b, 6c, 7a and 7b are "superimposed". Data cannot be read from them.

- In the scope of Conslusions, I propose to extend the information with plans for the future of research.

- The article lacks affiliation - it should be completed.

Reviewer 2 Report

Purpose of the paper

The purpose of the paper is to examine the neural response of learners when exposed to "old" and "new" stimuli. The authors aimed to induce learning by the participants and examine how different learning techniques and exposure can engage memory encoding. The study's findings indicate that when engaged in deep memory encoding, the participants exhibited greater P600 amplitude, but it was only for the abstract figures. However, there were methodological issues regarding the study design and ERP analysis that need further refining.

Comments

  1. The introduction ordering of information presented is confusing to follow. Furthermore, the purpose of the study is buried and lost in the introduction.
  2. How does LOP correlate with ERP response? The link between the theory and the measure is not well defined.
  3. Within the methods section, the authors indicate that abstract figures participants were asked to draw it, while non-words participants had to mention a rhyming word verbally. The authors indicate it is to induce deep and intermediate LOP. However, this is a major flaw as the study becomes unbalanced. The type of underlying neural processes is now different. Therefore, any contrast between the stimulus category will be skewed. Based on the theory presented in the introduction, the participants should learn and respond more accurately to the abstract figures than the non-words.
  4. ERP analysis, the authors have examined multiple ERP components N200, P300, N400, and P600. This type of analysis can cause ambiguities in the result interpretation. The authors should focus more on the specific components relevant to the paper's main research question. Currently, the paper feels like a shotgun approach and trying to see which components yield significance. Additionally, mathematical steps and calculations of the grand average waveform to extract the appropriate ERP component windows, "visual inspection" is not an appropriate method. The authors should consult the following papers on ERP.
    1. Luck, S. J. (2005). Ten simple rules for designing ERP experiments. Event-related potentials: A methods handbook262083337.
    2. Luck, S. J., & Gaspelin, N. (2017). How to get statistically significant effects in any ERP experiment (and why you shouldn't). Psychophysiology54(1), 146-157.

  1. The results sections were difficult to follow, as the tables and figures associated with the text were not placed together.
  2. Figure 3 does not accurately report the observed results section. Is the author trying to indicate the main effect for stimulus type or stimulus category?
  3. Figure 5 a. What interactions are the authors trying to display? It is currently unclear what the significance is indicating. Significant from other stimulus types or the regions? For example, "studied posterior" indicates "**" but what is the significance comparison? The frontal region? Or to another stimulus type. Similar confusion is illustrated within figure 5c, 5d, 7a, & 7b.

Round 2

Reviewer 1 Report

Dear Authors, 

Thank you for your replies. Most of the time they are satisfactory to me.

I already have the last comments for work:

- In terms of reference to the proposed articles, please kindly review, usually Authors reply with some delay. As a person with experience in this field - I always recommend contacting the author via e-mail. I believe that a short reference to the proposed titles of the publication would significantly complement the publication.

- Please include references to formulas / mathematical notations (1) and (2) in the text.

- I propose to enlarge 3A and 3B graphics, they are hardly visible in their present form.

- I am thinking of reorganizing points. 3.2 for the EEG, where we have subsection 3.2.1, then N200, P300 etc. are listed without the subsections. I don't know if the EEG subtitle is correct here, especially since we already have an EEG above in the article. Please rethink / correct it.

Reviewer 2 Report

The author's comments and revisions to the paper have been sufficient in addressing the concerns from the previous review. 

Author Response

Dear Reviewer 2,

We thank you once again for your time and constructive feedback. Your efforts are highly appreciated. We hope that you will find the final adjustments of our paper satisfactory.

Yours sincerely,

M. Toth, A. Sambeth, A. Blokland